# Target Detection Method for High-Frequency Surface Wave Radar RD Spectrum Based on (VI)CFAR-CNN and Dual-Detection Maps Fusion Compensation

**Yuanzheng Ji**, **Aijun Liu** *, **Xuekun Chen**, **Jiaqi Wang and Changjun Yu**

School of Information Science and Engineering, Harbin Institute of Technology at Weihai, Weihai 264209, China; 22S130417@stu.hit.edu.cn (Y.J.); 21b905055@stu.hit.edu.cn (X.C.); 21b905018@stu.hit.edu.cn (J.W.); yuchangjun@hit.edu.cn (C.Y.)
* Correspondence: liuaijun@hit.edu.cn

**Abstract:** This paper proposes a method for the intelligent detection of high-frequency surface wave radar (HFSWR) targets. This method cascades the adaptive constant false alarm (CFAR) detector variability index (VI) with the convolutional neural network (CNN) to form a cascade detector (VI)CFAR-CNN. First, the (VI)CFAR algorithm is used for the first-level detection of the range–Doppler (RD) spectrum; based on this result, the two-dimensional window slice data are extracted using the window with the position of the target on the RD spectrum as the center, and input into the CNN model to carry out further target and clutter identification. When the detection rate of the detector reaches a certain level and cannot be further improved due to the convergence of the CNN model, this paper uses a dual-detection maps fusion method to compensate for the loss of detection performance. First, the optimized parameters are used to perform the weighted fusion of the dual-detection maps, and then, the connected components in the fused detection map are further processed to achieve an independent (VI)CFAR to compensate for the (VI)CFAR-CNN detection results. Due to the difficulty in obtaining HFSWR data that include comprehensive and accurate target truth values, this paper adopts a method of embedding targets into the measured background to construct the RD spectrum dataset for HFSWR. At the same time, the proposed method is compared with various other methods to demonstrate its superiority. Additionally, a small amount of automatic identification system (AIS) and radar correlation data are used to verify the effectiveness and feasibility of this method on completely measured HFSWR data.

**Keywords:** HFSWR; cascade detection of CFAR-CNN; intelligent target detection; dual-detection maps fusion

## 1. Introduction

High-frequency surface wave radar (HFSWR) is a radar used for the over-the-horizon detection of the sea in medium- and high-frequency bands [1]. However, the radar echo signal contains components such as ground clutter, sea clutter, ionospheric clutter, and various sources of noise interference [2]. Among these, ionospheric clutter is a strong energy clutter caused by the reflection of radar signals from the ionosphere, while sea clutter is a frequency-symmetric first-order Bragg peak resulting from the reflection of electromagnetic waves by sea waves. The presence of clutter and noise makes it challenging to detect targets effectively using traditional single-frame detection methods, leading to false alarms or missed alarms, which raises the challenge to subsequent moving vessel tracking procedure [3].

The primary task of radar target detection is to extract useful target information from echoes while suppressing the interference of noise and clutter as much as possible [4]. The constant false alarm rate (CFAR) detection technique is one of the most classical radar detection techniques at present. In 1968, Finn et al. [5] proposed the cell averaging (CA-CFAR)

detection algorithm. Based on this, the smallest of (SO-CFAR) detection algorithm and the greatest of (GO-CFAR) detection algorithm were subsequently proposed. Later, in order to reduce the significant impact of strong interference from the detection background on clutter estimation, Ro ing et al. [6] proposed the Ordered Statistics (OS-CFAR) detection algorithm in 1983. Mean-based CFAR methods have a common characteristic of using averaging to estimate the local interference power level. The most classic method is CA-CFAR, followed by the development of methods such as SO-CFAR and GO-CFAR to improve the detection performance in non-uniform backgrounds. Among the ordered-based CFAR methods, OS-CFAR stands out as it utilizes ordered samples within a reference sliding window for processing. However, in a uniform background, the performance of ordered-based methods tends to be lower compared to mean-based methods (pp. 34–72, [7]). In 1986, Barnum [8] was the first to use HFSWR to detect the sea surface targets. The basic method of CFAR processing is to reference the unit samples and select samples that have consistent clutter characteristics with the detection unit background. Then, the clutter background power level is estimated to form a detection threshold. Therefore, background clutter identification and intelligent processing have become important research areas [9].

In recent years, many researchers have utilized machine learning techniques in the domain of intelligent radar target detection. In fact, research on machine learning in radar signal processing has predominantly focused on target recognition [10]. Deep learning models, like those used for SAR or ISAR images, map input data (such as SAR images, detailed range profiles, moving target micro-Doppler features, etc.) to specific output categories. This process is a form of nonlinear mapping. Target detection in radar operates similarly, using nonlinear mapping to identify a target's presence. In the 1990s, Professor Simon Haykin from McMaster University in Canada was the first to apply machine learning methods to clutter classification in radar target detection [11]. Building upon the work of Professor Haykin's team, researchers utilized MLP (multi-layer perceptron) and radial basis function networks for clutter classification based on single-frame or dual-frame observations [12]. In 2006, Professor Haykin [13] proposed the concept of cognitive radar, which consists of two main features: knowledge-assisted signal processing and adaptive transmission processing. In 2007, Besson [14] conducted research on a knowledge-assisted Bayesian detection framework under a non-uniform background and proposed a knowledge-assisted Bayesian detector. In 2012, Li Qinghua [15] proposed a knowledge-assisted CFAR detection algorithm. This approach effectively addresses the adverse effects caused by edge effects in the detection process.

In the domain of the radar signal processing, numerous scholars have applied deep learning methods to high-resolution radar target detection. However, for the HFSWRs that belong to short-wave radars, if the entire range–Doppler (RD) spectrum is used as input and machine learning networks are employed for target detection, the information of a single target only occupies a small fraction of the entire RD spectrum information. This situation poses greater challenges for machine learning networks in the processes of target recognition and localization. In the case of low-resolution radar, researchers are more concerned about integrating traditional signal detection or image processing techniques with deep learning methods. In 2019, Li Wang et al. [16] analyzed the potential application of deep neural networks (DNNs) in radar target detection. In 2020, Zhang Wandong [17] proposed a HFSWR target detection method based on optimal error self-correcting extreme learning machine (OES-ELM). In 2020, Li Qingzhong [18] proposed a fast sea surface target detection algorithm based on cascaded classifiers. In 2021, Wu et al. [2] proposed an intelligent object detection algorithm based on deep feature fusion. The aforementioned work can be summarized into a basic framework. Firstly, signal processing methods or image techniques are employed to perform the initial detection on the RD spectrum. Subsequently, machine learning or deep learning methods are utilized to achieve further classification. However, several issues still need to be explored and resolved in this process:

1.  The process of detecting the potential target regions on the RD spectrum using various signal processing or image processing methods lacks the flexibility of control-

ling preset false alarm probability parameters based on specific operator needs, as CFAR methods do, to achieve a balance between detection rate, false alarm rate, and miss rate.

2. In the detection process, the first-level detection method has a significant impact on the model training process. Therefore, it is necessary to develop a comprehensive model training plan.

3. Regardless of the classification model chosen, there is an objective problem of limited generalization ability, making it impossible to achieve the absolute and accurate recognition of targets or clutter. For example, when the signal-to-clutter ratio is low, the model may incorrectly identify some targets as clutter, resulting in a limited detection rate.

In conclusion, this paper proposes a pioneering detection model that combines the CFAR detector with a convolutional neural network (CNN), integrating image processing and computer vision techniques with CFAR processing. To address the issue of a limited detection rate caused by the convergence of the CNN network in the detection model, an innovative dual-detection map fusion method is introduced to compensate for the loss of detection performance. In brief, the results of an independent CFAR detector are utilized to compensate for the results of the CFAR-CNN cascaded detector. The present study introduces an innovative method that provides new ideas and solutions to solve the challenge of an HFSWR target detection. Empirical evidence from experiments confirms that the proposed method holds notable application value. Importantly, our method has the potential to provide a fresh research perspective for target detection tasks in other sensors. For instance, it could be applied to underwater target detection in the field of side-scan sonar, which offers high-resolution images [19,20]. We believe that our approach is applicable in this context, and such an extension can bring new perspectives and possibilities for target detection research in the domain of remote sensing.

## 2. (VI)CFAR-CNN Detector Model

### 2.1. (VI)CFAR Detector

In radar systems, the false alarm probability refers to the probability that the radar misjudges that a target exists when no target exists. In order to ensure that the radar system has predictable and stable detection performance, designers usually try to have a constant false alarm probability. The detection processor with a constant false alarm probability is called the constant false alarm rate (CFAR) processor. The following is the derivation of the principle of the general model of CFAR detection [21].

Assuming that the interference noise is independently and identically distributed, the probability density function of the detection unit $x_i$ is

$$P_{x_i}(x_i) = \frac{1}{\beta^2} exp(-x_i/\beta^2) \tag{1}$$

where $\beta^2$ is the full power of the IQ channel signal. The detection threshold of the detection unit needs to be $\beta^2$ set according to the estimated value. The idea of CFAR detection is to estimate $\beta^2$ based on $N$ reference units around the detection unit. These $N$ reference units construct the observation vector $\mathbf{x}$, whose joint probability density function is:

$$
\begin{aligned}
P_x(\boldsymbol{x}) &= \prod_{i=1}^{N} P_{x_i}(x_i) \\
&= \frac{1}{\beta^{2N}} \prod_{i=1}^{N} \exp\left(-x_i/\beta^2\right) \\
&= \frac{1}{\beta^{2N}} \exp\left(-\sum_{i=1}^{N} x_i/\beta^2\right)
\end{aligned}
\tag{2}
$$

By considering the joint probability density function of **x** as the likelihood function for **x**, we can compute the maximum likelihood estimate of $\beta^2$ as $\hat{\beta}^2$:

$$\hat{\beta}^2 = \frac{1}{N} \sum_{i=1}^{N} x_i \tag{3}$$

The detection threshold to be set can be obtained by multiplying $\hat{\beta}^2$ by a threshold coefficient $\eta$. This leads to the expression of the detection threshold as:

$$\begin{aligned} T &= \eta \beta^2 \\ &= \frac{n}{N} \sum_{i=1}^{N} x_i \end{aligned} \tag{4}$$

The mathematical expectation expression of the false alarm probability is:

$$\begin{aligned} \bar{P}_{FA} &= \int_{-\infty}^{\infty} e^{-\hat{T}/\beta^2} p_{\hat{T}}(\hat{T}) d\hat{T} \\ &= \left(\frac{N}{\alpha\beta^2}\right)^N \frac{1}{(N-1)!} \int_{-\infty}^{\infty} \hat{T}^{N-1} \exp\left\{-[(N/\alpha)+1]\hat{T}/\beta^2\right\} d\hat{T} \\ &= \left(1 + \frac{\alpha}{N}\right)^{-N} \end{aligned} \tag{5}$$

By presetting the expected false alarm probability, the expression of the threshold coefficient $\eta$ is obtained:

$$\eta = N(\bar{P}_{FA}^{-1/N} - 1) \tag{6}$$

VI (variability index)-CFAR detector, proposed by Michael and Pramod [22], is an adaptive detection method that combines CA-CFAR, SO-CFAR, and GO-CFAR. This method can dynamically adjust the estimation method for clutter power level by calculating the change index of the reference unit and the ratio of the front and rear sliding window mean. Previous studies (pp. 151–153, [7]) have shown that VI-CFAR has a lower complexity compared to OS-CFAR and a better false alarm performance compared to other adaptive detectors such as adaptive orders statistic (AOS)-CFAR [23] and estimation test (ET)-CFAR [24] detectors. We perform slow-time Fourier transform processing on the radar echo signal at each range gate to obtain the RD spectrum, where the range direction represents the radial distance of the target relative to the radar, and the Doppler spectrum represents the Doppler frequency shift of the target relative to the radar [8]. This article uses two-dimensional VI-CFAR processing, and its structural block diagram is shown in Figure 1.

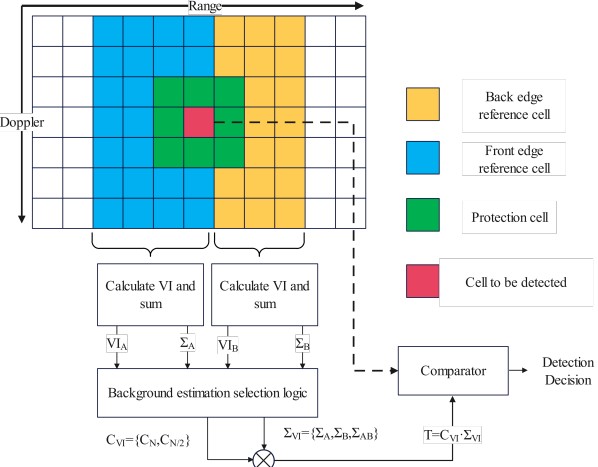

**Figure 1.** Two-dimensional VI-CFAR structural block diagram.

$VI$ is a second-order statistic, for the leading-edge reference cells and the trailing edge reference cells:

$$VI = 1 + \frac{\widehat{\sigma}^2}{\widehat{\mu}^2} = 1 + \frac{1}{N} \cdot \sum_{i=1}^{N} (x_i - \overline{x})^2 / (\overline{x})^2 \tag{7}$$

where $\widehat{\sigma}^2$ represents the estimated value of the variance, $\widehat{\mu}^2$ represents the estimated value of the mean square, and $\overline{x}$ represents the arithmetic mean of $N$ reference cells. By comparing $VI$ with the threshold $KVI$:

$$\begin{cases} VI \leq KVI \Rightarrow \text{Nonvariable} \\ VI > KVT \Rightarrow \text{Variable} \end{cases} \tag{8}$$

we can determine whether $VI$ comes from uniform or non-uniform clutter.

Define the mean ratio $MR$ of the leading edge reference cells and the trailing edge reference cells as [25]:

$$MR = \overline{x}_A / \overline{x}_B = \sum_{i \in A} x_i / \sum_{i \in B} x_i \tag{9}$$

By comparing $MR$ with threshold $KMR$:

$$\begin{cases} KMR^{-1} \leq MR \leq KMR \Rightarrow \text{ Same Means} \\ MR < KMR^{-1} \text{ or } MR > KMR \Rightarrow \text{ Different Means} \end{cases} \tag{10}$$

It can be judged whether the mean values of the front and back reference cells are the same. The VI-CFAR adaptive threshold is determined based on whether the aforementioned leading and trailing edge reference units are uniform and have the same mean value. The threshold coefficient $C_{VI}$ is determined based on the number of selected reference cells. When all current trailing edge reference units are used, the coefficient $C_N$ is used; when only the leading edge or trailing edge reference cells are used, the coefficient $C_{N/2}$ is used. According to Equation (6), the following expressions can be obtained:

$$C_N = \overline{P}_{FA}^{-1/N} - 1 \tag{11}$$

$$C_{N/2} = \overline{P}_{FA}^{-2/N} - 1 \tag{12}$$

The VI-CFAR adaptive threshold generation method is shown in the Table 1.

The VI-CFAR detector combines the advantages of CA, GO, and SO-CFAR. It is suitable for non-uniform backgrounds with more interference and clutter edges, such as HFSWR, and has certain robustness.

**Table 1.** VI-CFAR adaptive threshold generation method.

| Is the Leading Edge Reference Unit Uniform? | Is the Trailing Edge Reference Unit Uniform? | Are the Means the Same? | Adaptive Threshold | Equivalent Method |
|---|---|---|---|---|
| no | no | no | $C_N \sum\limits_{A\&B}$ | CA-CFAR |
| no | no | yes | $C_{N/2} max(\sum\limits_{A}, \sum\limits_{B})$ | GO-CFAR |
| yes | no | - | $C_{N/2} \sum\limits_{B}$ | CA-CFAR |
| no | yes | - | $C_{N/2} \sum\limits_{A}$ | CA-CFAR |
| yes | yes | - | $C_{N/2} min(\sum\limits_{A}, \sum\limits_{B})$ | SO-CFAR |

## 2.2. Clutter and Targets Classification by CNN

During the radar detection process, the visual processing abilities of experienced radar operators often outperform many CFAR processing methods in terms of the detection

of targets and management of false alarms. This can be attributed to the intricate visual observation mechanism of humans and their exceptional ability to extract image information. The effectiveness achieved in this manner cannot be replicated by any single CFAR method alone. Therefore, the integration of image processing techniques and computer vision technology with CFAR processing holds immense research value and has promising application prospects.

### 2.2.1. CNN Model

Convolutional neural network (CNN) is a deep neural network featuring localized connections, weight sharing, and additional defining traits. It is widely used in computer vision and pattern recognition domains, spanning various applications. CNN draws inspiration from the biological receptive field mechanism, wherein neurons solely receive signals from the localized area they govern. In the visual nervous system, the receptive field of a neuron refers to a specific region on the retina. Only when this area is stimulated can the neuron be activated [26]. A standard convolutional network consists of a convolutional layer, pooling layer, and fully connected layer as its fundamental components. The network structure is depicted in Figure 2.

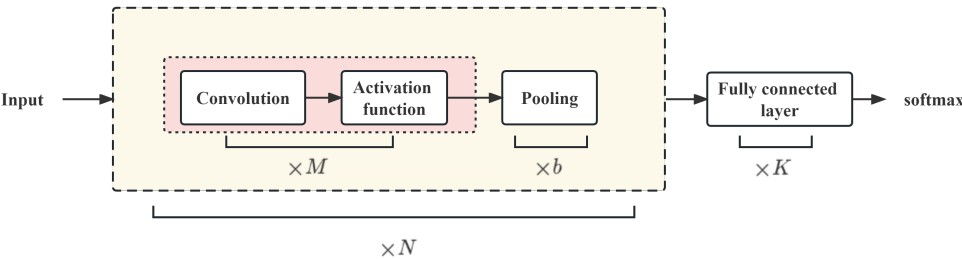

**Figure 2.** Basic structure of CNN.

In the CNN network, each convolutional block consists of $M$ convolutional layers and $b$ pooling layers. A CNN can be stacked by $N$ convolutional blocks, followed by $K$ fully connected layers. Convolution is the core of CNN. The input data enter the convolution layer after preprocessing. Assuming that the input data $\mathbf{X} \in \mathbb{R}^{M \times N}$ and a convolution kernel weight $\mathbf{W} \in \mathbb{R}^{U \times V}$ are given, the convolution of the input data $\mathbf{X}$ and the convolution kernel weight $\mathbf{W}$ are defined as:

$$\mathbf{Y} = \mathbf{W} * \mathbf{X} \tag{13}$$

where $*$ represents the convolution operation. The specific operation process is:

$$y_{ij} = \sum_{u=1}^{U} \sum_{w=1}^{V} w_{vs}.x_{i-u+1, j-v+1} \tag{14}$$

The primary function of the convolutional layer is to extract the features from a local region. By employing various convolutional kernels, it can perform equivalent operations as diverse feature extractors. The output of the convolutional layer can be represented by the following mathematical model:

$$\mathbf{Z} = f(\mathbf{W} * \mathbf{X} + b) \tag{15}$$

where $b$ represents the bias and $f(\cdot)$ represents the activation function. The activation function (activation layer) performs nonlinear operations on the data. In the CNN in this article, the relu function is employed as the activation function.

The output of the convolutional layer will enter the pooling layer. The function of the pooling layer is to decrease the dimensionality of the convolutional layer's output feature vector, thereby reducing the parameter count.

Assuming that the input of the pooling layer is divided into multiple regions, the pooling operation is to downsample each region to obtain a value as a summary of the region. Commonly used pooling functions include maximum pooling and average pooling. The CNN network in this article uses the maximum pooling method. Maximum pooling is defined as follows:

$$y_{m,n} = \max_{i \in R_{m,n}} x_i \tag{16}$$

The output of the convolutional block will be fed into the fully connected layer, which is a basic type of feed-forward neural network model known as the feedback neural network (FNN). The fully connected layer takes the features extracted by the convolutional block as its input and plays a role in classification throughout the entire CNN.

### 2.2.2. Experimental Data Acquisition

When the radar system receives multiple pulses, the range-pulse spectrum can be converted into a RD spectrum. The following formula is shown as:

$$\widehat{S}_r(t, \omega) = DFT(win \times \widehat{S}_r(t, k)) \tag{17}$$

where $\widehat{S}_r(t, k)$ represents the single-frame distance–pulse spectrum after pulse compression, $r$ represents the sequence number of the frame, $k$ represents the pulse number, and $t$ represents the fast time; $\widehat{S}_r(t, \omega)$ is the mathematical representation of the single-frame RD spectrum, where $\omega$ represents the Doppler frequency; the RD spectrum will be obtained by multiplying the window function *win* in the slow time domain and then performing discrete Fourier transform; here, we use the Hanning window as the window function to reduce the spectrum leakage and spectral peak side lobes, which helps to more accurately obtain the RD spectrum information of the target signal.

Figure 3 displays an RD spectrum after the above processing. The echo signals received by an HFSWR system located in the Bohai Sea are processed with two rounds of FFT to obtain the RD spectrum. The radar system parameters are shown in Table 2.

**Table 2.** HFSWR system parameters.

| Parameter | Value |
|---|---|
| Bandwidth | 50 kHz |
| Carrier frequency | 6.9 MHz |
| Time domain sampling rate | 400 MHz |
| Pulse width | $1.8 \times 10^{-4}$ s |
| Range resolution | 1.5 km |
| Doppler resolution | 0.004 Hz |

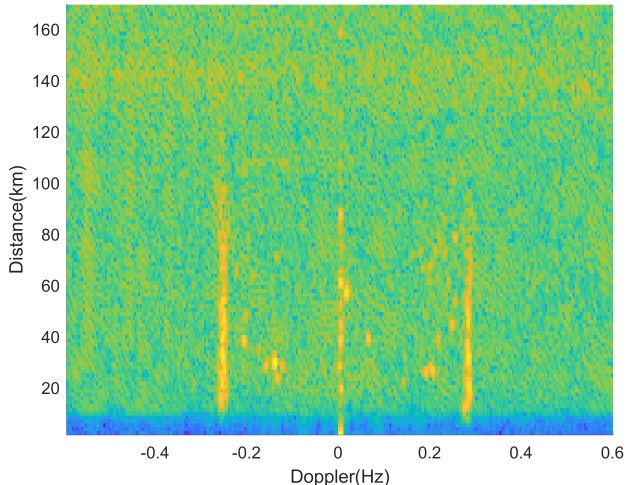

**Figure 3.** HFSWR measured RD spectrum.

In this spectrum, we can extract information related to the distance and Doppler frequency of the targets [27]. This spectrogram describes the power spectral density of electromagnetic wave signals received by HFSWR within the radar detection range. The color range from blue–green–yellow represents the gradual increase in echo energy. It can be seen from the spectrum that the detection environment of HFSWR is very complex. In the actual sea detection process of HFSWR, it is difficult to accurately and comprehensively obtain the true number, location, and status of sea surface targets. While it is feasible to use ship automatic identification system (AIS) data and radar echo data correlation as target true value information, vessels without AIS can still be detected by radar. However, if we only select targets with AIS data, this approach is not conducive to constructing an RD spectrum dataset that encompasses comprehensive position and velocity ground truth values. Moreover, it will also impact the training and testing of subsequent models. Therefore, in order to subsequently verify the performance of the proposed method and compare it with existing methods, this paper adopts the target embedding evaluation method—embedding a certain number of simulated targets in the clutter background to construct a real scene of real HFSWR targets [28]. In addition, we obtained a small amount of association data between the AIS and radar echoes of the target to verify the feasibility of detecting the target using the proposed method in measured target data.

In the original radar RD spectrum, we manually extracted the detection background. Firstly, a distance range of 20–70 km, and Doppler ranges of −0.5 Hz−−0.3 Hz and 0.3 Hz–0.5 Hz are selected from the fully measured RD spectrum as the background reference region and the distribution of this region is calculated. Then, the two regions between the positive and negative Bragg peaks and the zero frequency are randomly assigned values based on the distribution of the background reference region, resulting in a spectrum representing the background detection, as shown in Figure 4.

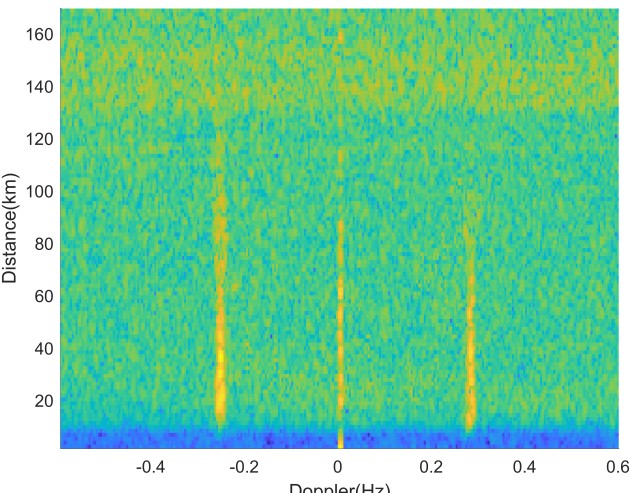

**Figure 4.** Measured detection background.

In the detection background, we add a certain number of simulated targets. For each coherent integration period, we set the number of targets and their specific distance and velocity information. We adopt a linear frequency interrupted continuous wave (LFICW) approach where $N_{cit}$ pulses are transmitted within a coherent integration period. Each pulse in the received signal is sampled $R_{num}$ times, and after performing a single fast Fourier transform (FFT), $R_{num}$ range bins are obtained. Assuming there are $m$ targets in a specific range gate, the signal model of the sum of $m$ target signals can be represented as:

$$s_m(kT) = A_i \sum_{i=1}^{m} exp(j\pi K(k(T - \tau_i))^2 + j2\pi f_i k(T - \tau_i)) \tag{18}$$

where $T$ represents the sampling interval, $k$ is the number of sampling points, $A_i$ and $f_i$, respectively, represent the amplitude and Doppler frequency of the ith target, $\tau_i$ represents the distance time delay of the ith target signal, $K$ represents the frequency modulation slope. Then, we obtained the RD data $\mathbf{T}_{\mathrm{rd}} \in \mathbb{C}^{R_{num} \times N_{cit}}$ of the simulation target according to Formula (17), where $R_{num}$ represents the number of range gates, and $N_{cit}$ represents the number of coherent accumulation cycles. The mathematical expression of $\mathbf{T}_{\mathrm{rd}}$ is as follows:

$$\mathbf{T}_{\mathrm{rd}} = \begin{bmatrix} t_{11} & t_{12} & \cdots & t_{1N_{\mathrm{cit}}} \\ t_{21} & t_{22} & \cdots & t_{2N_{\mathrm{cit}}} \\ \vdots & \vdots & \ddots & \vdots \\ t_{R_{\mathrm{num}}1} & t_{R_{\mathrm{num}}2} & \cdots & t_{R_{\mathrm{num}}N_{\mathrm{cit}}} \end{bmatrix} \tag{19}$$

Finally, we superimpose the simulated target RD data and the measured background data $\mathbf{B}_{\mathrm{rd}} \in \mathbb{C}^{R_{num} \times N_{cit}}$:

$$\mathbf{S}_{\mathrm{rd}} = \mathbf{T}_{\mathrm{rd}} + \mathbf{B}_{\mathrm{rd}} \tag{20}$$

In this way, we can obtain the RD spectrum $\mathbf{S}_{\mathrm{rd}}$ of the semi-measured and semi-simulated radar that contains all the target true value information. Figures 5 and 6 show a frame of RD spectrum containing 45 simulation targets and 35 simulation targets, respectively. For the convenience of viewing, the simulation targets are marked on the left figure and the simulation targets are not marked on the right figure.

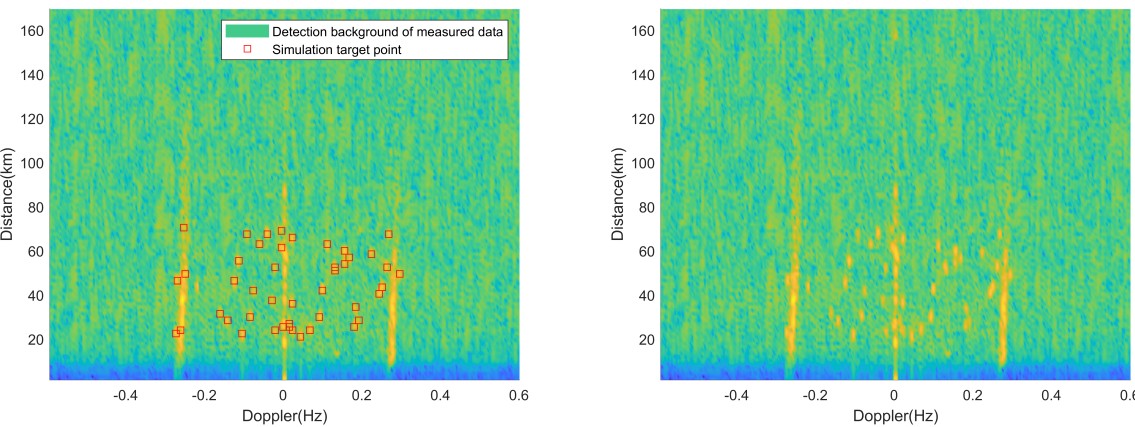

**Figure 5.** Single-frame RD spectrum containing 45 simulation targets.

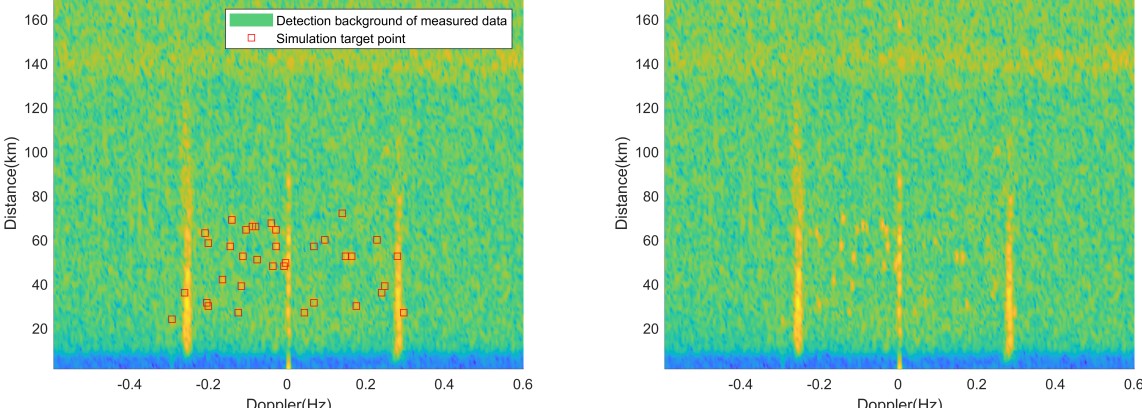

**Figure 6.** Single-frame RD spectrum containing 35 simulation targets.

It is worth mentioning that the simulated targets are within a distance range of 20–70 km. In practice, the HFSWR system has a detection blind zone of approximately

20 km. Additionally, different targets are located in different range and Doppler cells. The range resolution in the spectrum is 1.5 km and the Doppler resolution is 0.004 Hz. However, due to signal extensions in both the range and Doppler domains, two targets that are close in the spectrum can exhibit overlap, which is also observed in practical detection.

The HFSWR emits electromagnetic waves that interact with the rough sea surface. As a result, the resulting echo signal exhibits a ridge-like structure symmetrical about the Doppler zero frequency in the upward distance of the RD spectrum. Furthermore, the ionospheric clutter manifests as a band-like structure in the upward direction of the Doppler spectrum. Additionally, ground clutter is evident with a ridged structure similar to sea clutter at the Doppler zero frequency. In contrast, the target point in the RD spectrum appears as an isolated peak with a specific amplitude [29]. In fact, these descriptions of different types of clutter and targets on the RD spectrum come from the human observation and perception of different spectral structures, and CNN can automatically extract the different characteristics of targets and clutter and conduct a one-step comparison, achieving the end-to-end classification of clutter and targets.

We utilize (VI)CFAR for the preliminary detection to obtain the first-level detection result for a single frame of RD spectrum data. Using the position of this result in the RD spectrum as the center, a $9 \times 9$ rectangular window is extracted from the RD spectrum to form a window slice, which contains the target or clutter in the form of two-dimensional image data. In terms of window size selection, we take into account the range and Doppler extensions of the target in the spectrum. However, the number of extension cells in each direction is generally limited to four. This ensures that the window slice contains all the information about the target in the spectrum while avoiding the redundancy caused by larger window sizes that may incorporate interference from other targets or clutter. Subsequently, CNN is employed to classify the target or clutter. Figure 7 presents an example of window slices showing clutter and targets.

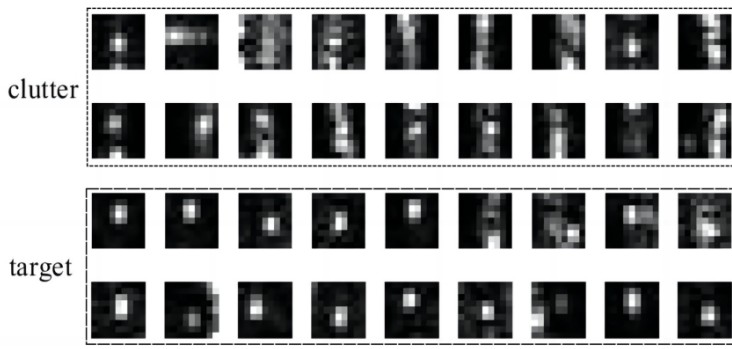

**Figure 7.** Example of window slicing between clutter and target.

### 2.2.3. Training and Testing Process of (VI)CFAR-CNN

When a deep learning network is introduced to achieve radar target detection, the training of the network model is crucial. A simple training strategy is to perform sliding window processing on each frame of the RD spectrum to form window slice data that can be input to the network. Since the distance and speed information of the simulated target are known, the "target" tag is only added when the target is located in the detection unit; otherwise, a "non-target" tag is added. The problem with this strategy is that if the signal-to-noise ratio (SNR) of the target is very low, even if more such samples are trained, it will be difficult for the network model to distinguish whether it is a target or a clutter during the detection process; in addition, an imbalanced ratio of positive examples (targets) to negative examples (clutter) within the frame's RD spectrum also has an impact on network training.

The training strategy of this article is to first use the CFAR detector to detect in each frame of the RD spectrum; the obtained detection results are centered on their position in the RD spectrum, and a $9 \times 9$ window is used to intercept them to form a window

slice datum; based on the known target distance and speed information of each frame of data, and the distance and speed information detected by CFAR contained in each window slice datum, we can compare these information and determine whether each window slice datum is a target. Using this approach, each window slice datum can be appropriately labeled as either a target or a non-target. This labeling process effectively utilizes the CFAR detector to supervise and guide the CNN network in distinguishing between targets and non-targets.

Table 3 shows the CNN network structure parameters in the proposed method. In the network, window slice data are used as network input; the convolution layer is responsible for extracting features and obtaining the spatial structure information in the window slice through convolution operations; a batch normalization layer (BN) is introduced between the layers and the pooling layer to optimize the training and bolster network stability; the output of the last pooling layer is flattened into a one-dimensional vector and sent to the fully connected layer to complete classification.

**Table 3.** CNN network structure parameters.

| Number of Layers | Layer Type | Size | Step | Channel Size |
|---|---|---|---|---|
| 1 | Convolution layer 1 | $3 \times 3$ | 1 | 32 |
| 2 | Pooling layer 1 | $2 \times 2$ | 2 | 32 |
| 3 | Convolutional layer 2 | $3 \times 3$ | 1 | 64 |
| 4 | Pooling layer 2 | $2 \times 2$ | 2 | 64 |
| 5 | Convolutional layer 3 | $3 \times 3$ | 1 | 128 |
| 6 | Pooling layer 3 | $2 \times 2$ | 2 | 128 |
| 7 | Convolutional layer 4 | $3 \times 3$ | 1 | 64 |
| 8 | Pooling layer 4 | $2 \times 2$ | 2 | 64 |
| 9 | Fully connected layer 1 | - | - | 32 |
| 10 | Fully connected layer 2 | - | - | 16 |
| 11 | Fully connected layer 3 | - | - | 2 |

Note: The activation layer and batch normalization layer are omitted in the table.

Figure 8 shows the training and testing processes of (VI)CFAR-CNN detector.

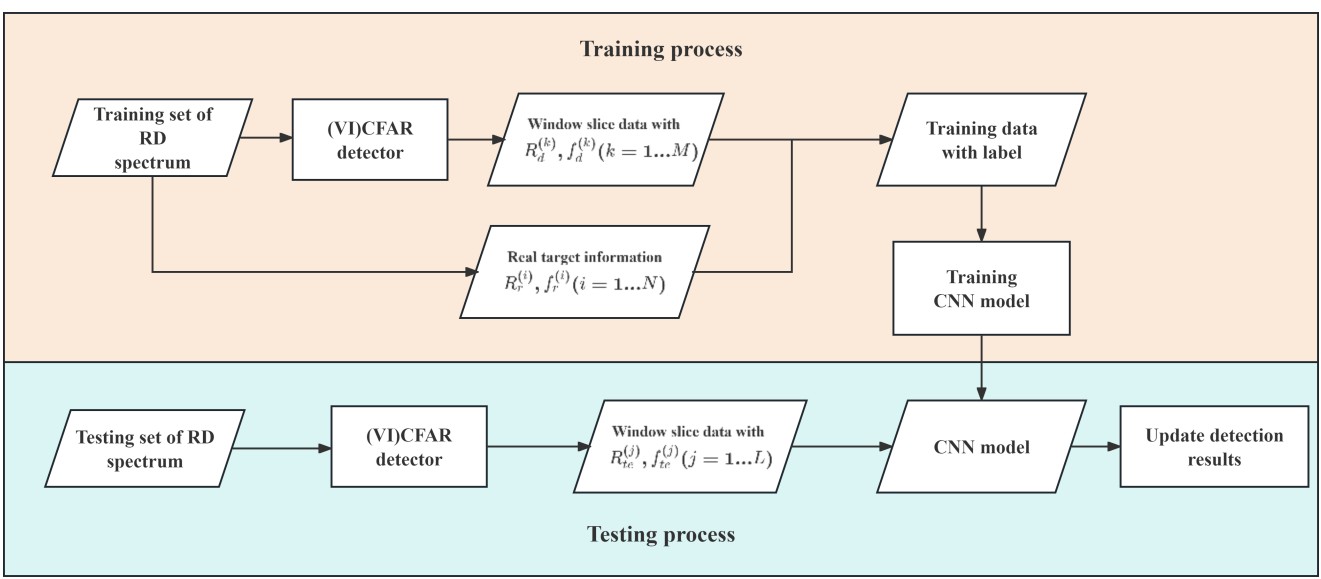

**Figure 8.** Training and testing process of (VI)CFAR-CNN.

During the training process, the CFAR detector detects "suspected targets" to form window slice data and records range and Doppler frequency information, and compares them with the actual range and Doppler information of the target, and if a match is found, add the label "target"; otherwise, add the label "non-target". The multi-frame simulation

constructs a training dataset consisting of window slices and their corresponding labels, which is then used for training the CNN model. During the test process, use CFAR for detection to obtain preliminary detection results, obtain window slice data based on the detection results, and then use the trained CNN model to further classify whether the window slice data are clutter or targets, and finally update the detection results to complete the detection.

In the (VI)CFAR-CNN detector, the training of the CNN model is supervised by the previous level CFAR detector. Different preset false alarm probability settings of the CFAR detector will produce different window slice data, such as using preset false alarms. In the window slice data generated by the CFAR detector with a low false alarm probability, the SNR of the detected target is generally high, and the proportion of clutter data in the overall data is small; while in the window slice data generated by the CFAR detector with a high preset false alarm probability, the range of the SNR of the detected target is larger, and the clutter data account for a larger proportion of the overall data. Therefore, different preset false alarm probability settings will affect the training of the CNN model. Here, we compare the results of training and testing on window slice data generated by CFAR detectors using different preset false alarm probabilities.

We used CFAR detectors with different preset false alarm probabilities (0.001, 0.002, 0.005, 0.01, 0.02, and 0.05) to generate different window slice data, and fused the window slice data generated by the above multiple preset false alarm probabilities to obtain a fused dataset. Then, we trained and tested different CNN models using the above seven types of datasets, and calculated four evaluation indicators: classification accuracy, recall, precision, and F1 score. For the sake of fairness, each type of dataset contains 2600 clutter window slices and 1000 target window slices, of which 1/10 is used for testing and the rest is used for training. According to the performance metrics shown in Table 4, we discovered that the CNN model trained using window slice data generated by the CFAR detector with multiple preset false alarm probabilities demonstrates a superior performance. Therefore, in subsequent experiments, we used window slice data generated by the CFAR detectors with multiple preset false alarm probabilities to train the CNN model.

**Table 4.** CNN model performance under different CFAR preset parameters.

| First Level CFAR Preset Parameters | Classification Accuracy | Recall | Precision | F1 Score |
|:---:|:---:|:---:|:---:|:---:|
| 0.001 | 0.9778 | 0.9457 | 0.9667 | 0.9560 |
| 0.002 | 0.9389 | 0.9383 | 0.8172 | 0.8736 |
| 0.005 | 0.9528 | 0.9222 | 0.8925 | 0.9071 |
| 0.01 | 0.9722 | 0.9667 | 0.9255 | 0.9457 |
| 0.02 | 0.9639 | 0.9608 | 0.9159 | 0.9378 |
| 0.05 | 0.9667 | 0.9307 | 0.9495 | 0.9400 |
| **Multiple PFA Parameters** | 0.9806 | 0.9899 | 0.9423 | 0.9655 |

## 3. Fusion of Dual-Detection Maps to Compensate for Detection Performance

### 3.1. Detection Rate Loss of (VI)CFAR-CNN

In the previous article, we introduced the principle and process of (VI)CFAR-CNN detection, in which the detection rate of the detector is limited due to the convergence of the CNN model. As shown in Figure 9, we plotted the ROC curves of (VI) CFAR and (VI) CFAR-CNN, illustrating the correlation between detection rate (DR) and false alarm rate (FAR). Their calculation formula is as follows:

$$DR = N_c / N_{total} \tag{21}$$

$$FAR = N_{far} / N_{rd} \tag{22}$$

where $N_c$ represents the number of correctly detected targets, $N_{total}$ represents the total number of targets in the frame spectrum, $N_{far}$ represents the number of false alarm units, and $N_{rd}$ represents the total number of intercepted RD spectrum units (in this article it is

$113 \times 299$). All performance indicators related to DR and FAR in this article are calculated according to the above formula.

We can find that, due to the cascaded CNN model after (VI)CFAR, the detection performance of (VI) CFAR-CNN is significantly better than (VI)CFAR. However, due to the convergence of the CNN model, the detector output results tend to be stable. After stabilizing, the DR of (VI)CFAR-CNN no longer increase when it reaches 0.9592, which is called the detection rate loss. The DR and FAR of (VI)CFAR have always kept changing synchronously, and there is no loss of DR. According to the Neyman–Pearson criterion, we hope to control the false alarm probability within a specified range and increase the detection probability as much as possible. Obviously, the (VI)CFAR-CNN detection rate loss caused by the convergence of the CNN model is inconsistent with this criterion. When the model convergence point is reached, the DR will not increase even if the preset false alarm probability is increased. Therefore, we need a way to compensate for the (VI)CFAR-CNN detection rate loss.

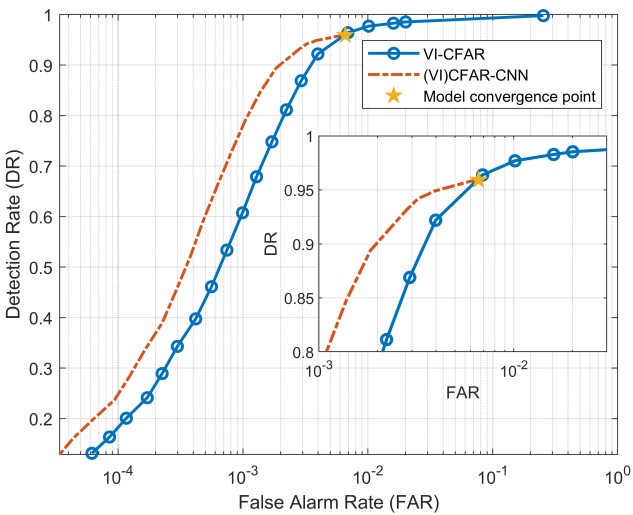

**Figure 9.** ROC of (VI)CFAR-CNN and (VI)CFAR.

*3.2. Method of Fusion Decision Making of Dual-Detection Maps*

This paper proposes a methodology for making comprehensive decisions based on dual-detection maps in order to enhance the detection performance.

First, the (VI)CFAR-CNN detector with a preset false alarm probability of $P_{con}$ is used for detection on the RD spectrum ($P_{con}$ is the preset false alarm probability when the detector model converges), and the detection map A is obtained.

Then, use the (VI)CFAR detector with the preset false alarm probability of $P_B$ to detect ($P_{con} \leqslant P_B$), and the detection map B is obtained.

For the detection map A and the detection map B, we regard them as two matrices **detecA** and **detecB**, and **detecA** $\in \{0,1\}^{m \times n}$, **detecB** $\in \{0,1\}^{m \times n}$. The value "1" is used to indicate that the target is detected, while the value "0" indicates that the target is not detected. Subsequently, the detection results are weighted and fused using the following calculation method:

$$\mathbf{detecC} = \begin{bmatrix} \omega_A & \omega_B \end{bmatrix} \begin{bmatrix} \mathbf{detecA} \\ \mathbf{detecB} \end{bmatrix} \tag{23}$$

where $\boldsymbol{\omega} = \begin{bmatrix} \omega_A & \omega_B \end{bmatrix}$ represents the weight vector of the two detectors, and $\boldsymbol{\omega} \in (0,1)^{1 \times 2}$ and $\omega = 0.d$, where $d \in \{1,2,3,\ldots,9\}$ represents the detection matrix after weighted fusion. For each connected component in **detecC**, perform the following calculations:

$$D_k = \frac{\sum_{i=1}^{N_k} \xi_{k,i}}{\mu N_k} \tag{24}$$

where $N_k$ represents the quantity of units within the k-th connected component; $\xi_{k,i}$ represents the value of the i-th unit in the k-th connected component; $\mu \in (0,1)$ and $\mu = 0.d$, where $d \in \{1, 2, 3, \ldots, 9\}$ is the normalized factor of the connected component; for parameters $D_k$, we make the following judgments and perform the following operations:

$$\forall i, \quad \xi_{k,i} = \begin{cases} 1, & D_k > 1 \\ 0, & D_k \leqslant 1 \end{cases} \tag{25}$$

After traversing the connected components and completing the calculations mentioned above, the final detection results of the dual-detection maps fusion decision are obtained.

In fact, we use an independent (VI)CFAR detector to compensate for the loss of detection performance due to the convergence of the (VI)CFAR-CNN model, ensuring that the preset false alarm probability of (VI)CFAR-CNN remains unchanged, and the preset false alarm probability of (VI)CFAR is increased, thereby improving the final DR, but it will also increase the FAR. As previously stated, the DR and FAR of the detector exhibit synchronous increments or decrements. Nevertheless, our objective for improvement is to maximize the DR while maintaining a FAR that falls within an acceptable threshold.

### 3.3. Optimization of the Weight and Normalized Factor of the Double Detection Map

In the aforementioned process of fusion decision-making of dual-detection maps to compensate for detection performance, the values of the weights $\omega_A$ and $\omega_B$ of the dual-detection maps and the normalized factor $\mu$ are crucial to the quality of the final fusion result.

Under the preset false alarm probability $P_B$ of each independent (VI)CFAR, the value strategy of $\omega_A$, $\omega_B$ and $\mu$ becomes an optimization problem. For this optimization problem, we use the following mathematical model to describe:

$$\begin{cases} \min f(\boldsymbol{x}) \\ \text{s.t. } \boldsymbol{x} = [\omega_A, \omega_B, \mu]^T, \text{ where } \omega_A, \omega_B, \mu \text{ are one-digit decimals from 0.1 to 0.9} \end{cases} \tag{26}$$

where $f$ represents the objective function, x represents the vector composed $\omega_A$ of $\omega_B$ and $\mu$.

We let $\Omega = \{p_1, p_2, \cdots, p_n\}$, and $\forall p_i \in \Omega, p_i \geq P_{con}$, where there is the parameter space of the preset false alarm probability of all (VI)CFAR in the dual-detection maps fusion decision algorithm. For each element in the parameter space, there exists an optimal solution $\boldsymbol{x}^*$ obtained through the corresponding optimization algorithm. Subsequently, we present the specific optimization process.

We use pseudocode to describe the definition of the objective function. The calculation process of the objective function is shown in Algorithm 1. In this process, the objective function inputs $\omega_A$, $\omega_B$, and $\mu$; and then we load a fixed RD spectrum dataset; and then we use the dual-detection maps fusion decision-making method to obtain the detection results of each frame of the RD spectrum, and calculate the overall DR and FAR; finally, the objective function value objective about the DR and FAR is output. It should be noted that $p_{fa1}$ and $p_{fa2}$ in the objective function are global variables outside the function. Let $p_{fa1}$ be constant as $P_{con}$ and only change $p_{fa2}$ and $p_{fa2} \in \Omega$ as needed.

We use the Surrogateopt algorithm to optimize $\omega_A$, $\omega_B$, and $\mu$. The definition of the optimization function is shown in the pseudocode in Algorithm 2. The Surrogateopt algorithm is an optimization algorithm based on a surrogate model. It is suitable for objective functions that have a time-consuming and laborious calculation process.

---

**Algorithm 1:** Objective function

**Input:** Weight of detectionMap1: $\omega_A$, Weight of detectionMap2: $\omega_B$, Normalized factor: $\mu$

1   Load RD spectrum dataset;

2   $p_{fa1}$: The preset expected false alarm probability of the (VI)CFAR-CNN detector;

3   $p_{fa2}$: The preset expected false alarm probability of the VI-CFAR detector;

4   *frameNum*: The number of frames in the RD spectrum dataset;

5   $PD_{total} \leftarrow 0$;

6   $PFA_{total} \leftarrow 0$;

7   **for** *each frame in frames* **do**

8     Use (VI)CFAR-CNN detector with preset expected false alarm probability $p_{fa1}$ to detect the RD spectrum of the current frame, and detectionMap1 was obtained;

9     Use VI-CFAR detector with preset expected false alarm probability $p_{fa2}$ to detect the RD spectrum of the current frame, and detectionMap2 was obtained;

10     Detection map fusion is performed with detectionMap1 and detectionMap2 based on the value of $\omega_A$ $\omega_B$ and $\mu$;

11     $P_D \leftarrow$ Calculate the detection rate of the fusion detection map;

12     $P_{FA} \leftarrow$ Calculate the false alarm rate of the fusion detection map;

13     $PD_{total} \leftarrow PD_{total} + P_D / frameNum$;

14     $PFA_{total} \leftarrow PFA_{total} + P_{FA} / frameNum$;

15   **end**

16   $objective \leftarrow (1 - PD_{total})^2 + PFA_{total}^2$;

17   **return** *objective*

---

**Algorithm 2:** Optimization process of $\omega_1$, $\omega_2$ and $\mu$

**Input:** Objective function: $f$, Iterations: *evals*

1   construct surrogate;

2   $solution \leftarrow \varnothing$;

3   $currentVal \leftarrow \infty$;

4   $iter \leftarrow 0$;

5   *point* : a vector consisting of three variables $\omega_1$, $\omega_2$, and $\mu$ ;

6   **while** *iter < evals* **do**

7     Compute surrogate model;

8     Choose between enhancing the surrogate model or identifying the optimal point using the existing one;

9     *point* $\leftarrow$ Identify the subsequent point for consideration based on the preceding decision;

10     **if** $f(point) < currentVal$ **then**

11       $solution \leftarrow point$;

12       $currentVal \leftarrow f(point)$;

13     **end**

14     $iter \leftarrow iter + 1$;

15   **end**

16   **return** *solution*

---

The algorithm approximates the actual objective function by constructing a surrogate model and reduces the number of solutions to the actual objective function [30]. In each iteration, the Surrogateopt algorithm selects a suitable point to evaluate the actual objective function and updates the surrogate model based on the evaluation results. Through contin-

uous iteration, the Surrogateopt algorithm can find the optimal solution or a solution close to the optimal solution.

After the aforementioned optimization process, each preset parameter (i.e., the preset false alarm probability of (VI)CFAR) corresponds to an optimal combination of weight and normalized factor. During the implementation of the detection, we can configure these parameters using a lookup table based on the preset values.

## 4. Results and Discussion

### 4.1. Introduction to Simulation Target Experimental Data

According to the experimental data acquisition method described in the above section, we constructed two sets of HFSWR target-embedded RD spectrum data to evaluate the proposed method. The two sets of datasets, DATA_1 and DATA_2, are represented in Table 5.

**Table 5.** Dataset Information.

| Dataset | Frames | Single Frame Target Number | Average SNR (dB) | Average SCR (dB) | Number of Frames Used for Training | Optimize the Number of Frames Used | Number of Frames Used for Testing |
|---|---|---|---|---|---|---|---|
| DATA_1 | 130 | 45 | 22.4963 | 5.0563 | 30 | 20 | 80 |
| DATA_2 | 130 | 35 | 20.2041 | −0.4026 | 30 | 20 | 80 |

As shown in Figures 10 and 11, this illustrates the relative intensity information of the RD spectrum targets, clutter, and noise for 80 frames tested in datasets Data_1 and Data_2. The three colors in the figure represent the target, background noise, and clutter, respectively. The "dots" represent the relative intensity mean, and the "upper and lower horizontal bars" represent the relative intensity range.

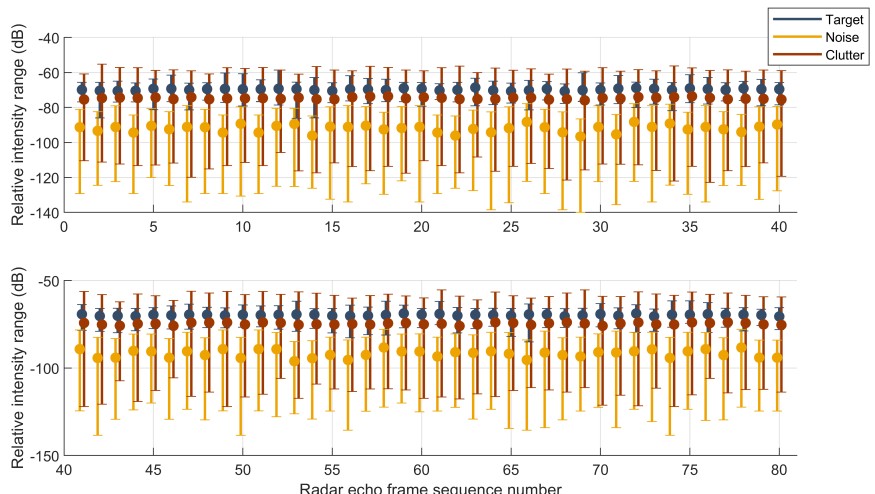

**Figure 10.** Relative intensity range of target, clutter and noise of DATA_1.

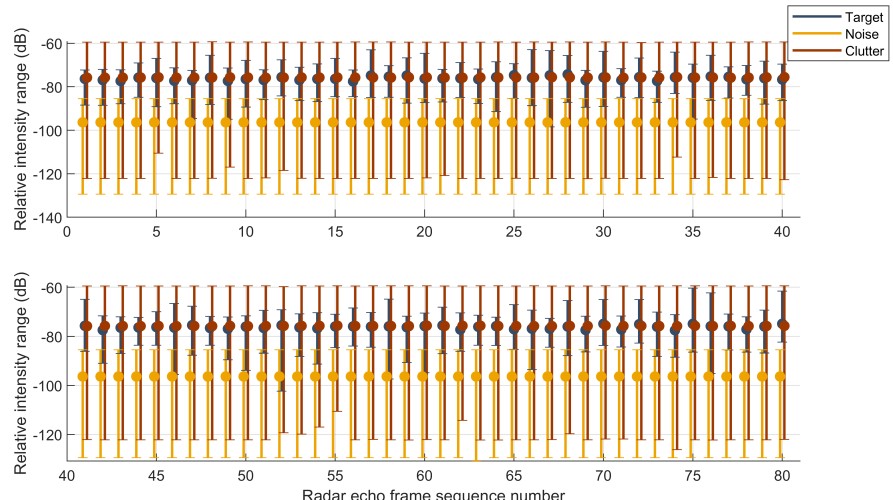

**Figure 11.** Relative intensity range of target, clutter, and noise of DATA_2.

## *4.2. Comparison of ROC Curves of Detection Methods*

The radar target detection receiver operating characteristic (ROC) curve can be utilized for evaluating and comparing various detection performances. It describes the relationship between DR and FAR in detection methods. The area under the curve (AUC, area under the curve) is usually used to evaluate detection methods, corresponds to their performance. A larger AUC indicates better performance.

### 4.2.1. Comparison of Different CFAR Detectors Used as First-Level Detection

This section aims to verify the detection performance of using different CFAR detectors for first-level detection and subsequently applying the same CNN model for second-level classification. Based on the RD spectrum data of DATA_1 and DATA_2, we compared three solutions that employ different CFAR as the first-level detection method, namely (VI)CFAR-CNN, (SO)CFAR-CNN, and (CA)CFAR-CNN. As shown in Figures 12 and 13, the experimental results show that (VI)CFAR-CNN has the most superior detection performance, while (SO)CFAR-CNN has equivalent performance to (CA)CFAR-CNN. Since (VI)CFAR is an adaptive CFAR method, it is more suitable for detecting high-frequency ground wave radar data with an uneven background. The experimental results align with the underlying theory, thereby substantiating the efficacy and applicability of (VI)CFAR as a first-level detection.

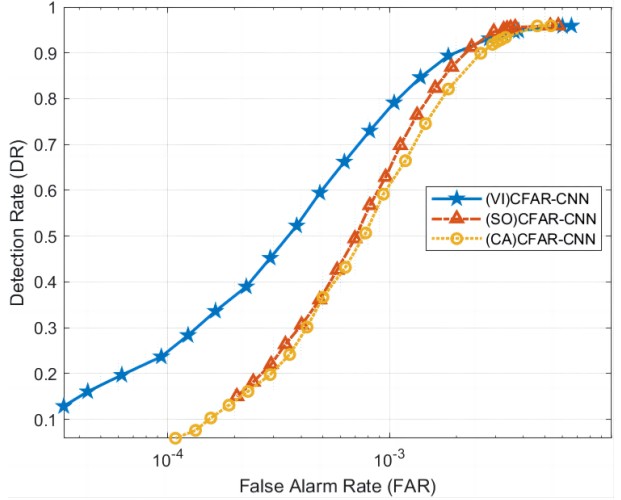

**Figure 12.** Comparison of the ROC curves using different CFAR methods as the first level of detection in DATA_1.

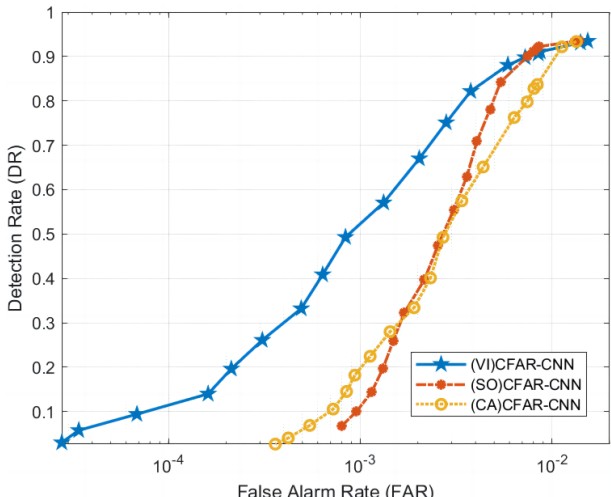

**Figure 13.** Comparison of ROC curves using different CFAR methods as the first level of detection in DATA_2.

However, regardless of the chosen CFAR as the first-level detection method, once the CNN model reaches the convergence state, there is no further improvement in the detection rate. The ROC curves of (VI)CFAR-CNN, (SO)CFAR-CNN, and (CA)CFAR-CNN gradually converge.

4.2.2. Detection Performance Compensation Effect of Dual-Detection Maps Fusion Decision-Making

To address the issue of limited DR resulting from the convergence of the CFAR-CNN model, this paper presents a novel approach called the dual-detection maps fusion compensation method. In this method, we fix the preset false alarm probability of the first detector (VI)CFAR-CNN, and need to set the preset false alarm probability of the second detector (VI)CFAR, as well as the weights of both detection maps and the value of the normalized factor. The fusion decision of the dual-detection maps compensates for detection loss.

The ROC curve after the dual-detection map fusion compensation is shown in Figures 14 and 15. It can be seen from the figures that the loss of detection performance due to model convergence has been compensated, and the DR has been improved. In addition, the FAR has also been suppressed to a certain extent, which is lower than (VI)CFAR.

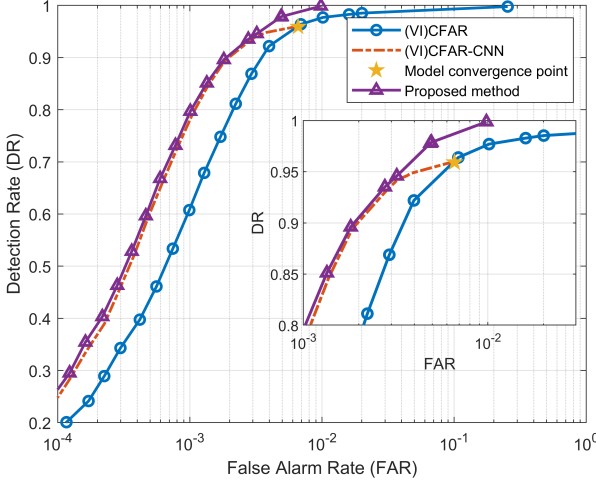

**Figure 14.** ROC curve after the fusion and compensation of dual-detection maps in DATA_1.

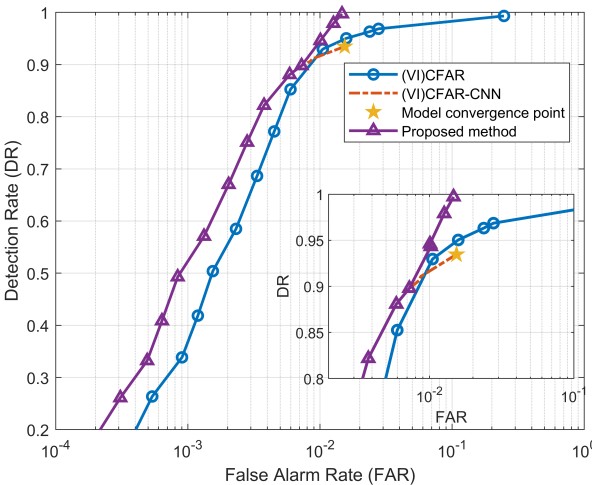

**Figure 15.** ROC curve after the fusion and compensation of dual-detection maps in DATA_2.

### 4.2.3. Comparison of ROC Curves between This Method and Other Methods

As shown in Figures 16 and 17, we use the proposed method with two CFAR-CNN detectors that use different CFAR methods as first-level detection (i.e., (SO)CFAR-CNN and (CA)CFAR-CNN), as well as three different CFAR detectors (i.e., (VI)CFAR, (CA)CFAR, and (SO)CFAR) for comparison. By comparing the performance of these methods on both datasets, several conclusions can be drawn.

Firstly, the obtained ROC curve clearly demonstrates the superior performance of the proposed method in this article, as indicated by its larger area under the curve. This result demonstrates our method's improved accuracy in target detection while notably reducing false alarms.

Secondly, despite DATA_2 having a relatively lower SNR and SCR compared to DATA_1, the proposed method exhibits strong detection performance on both datasets. This demonstrates the good robustness and adaptability of our method, showcasing its ability to yield consistent results even when handling low SNR or SCR datasets.

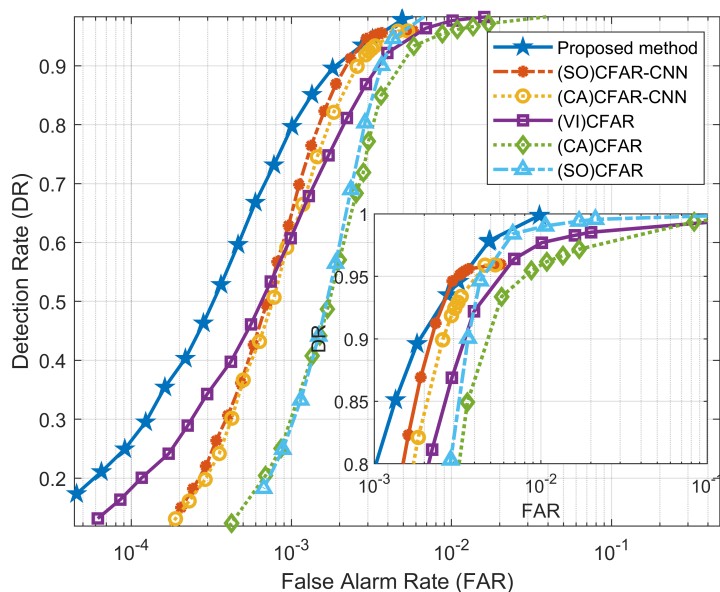

**Figure 16.** Comparison of ROC curves between proposed method and other methods in DATA_1.

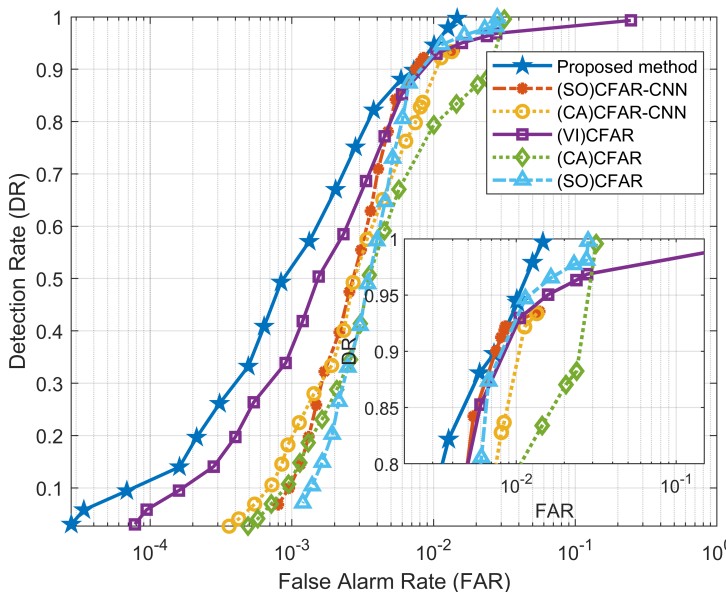

**Figure 17.** Comparison of ROC curves between the proposed method and other methods DATA_2.

*4.3. Comparison of RD Spectrum Detection Effects*

4.3.1. Comparison between the Proposed Method and Traditional CFAR Detection Method

This section presents a comparison between the method proposed in this article and the traditional CFAR detection method in order to evaluate their respective detection effectiveness. To illustrate the comparison results, a frame of RD spectrum data from DATA_1 and DATA_2 was selected as an example, and the detection effects were compared using both the proposed method and the traditional CFAR method.

As shown in Figure 18, the left column represents the detection effect using DATA_1 single-frame RD spectrum data, while the right column represents the detection effect using DATA_2 single-frame RD spectrum data. In each detection result figure, the yellow area indicates the detected target results, the blue area indicates the target not being detected, and the red box indicates the position of the actual target in the RD spectrum. This article utilizes two different detection methods, (VI)CFAR and (CA)CFAR, for a comparison with a proposed method. The preset parameters for each detection method are shown in Table 6:

**Table 6.** Parameter settings for each method.

| Data Sources | Detection Method | Parameter Settings |
|---|---|---|
| DATA_1 | Proposed method | (VI) CFAR-CNN, the preset false alarm probability is 0.036, KVI = 4.7, KMR = 1.05, without dual-detection maps fusion |
| | (VI)CFAR | The preset false alarm probability is 0.006, KVI = 4.7, KMR = 1.05 |
| | (CA)CFAR | The preset false alarm probability is 0.017 |
| DATA_2 | Proposed method | (VI) CFAR-CNN, the preset false alarm probability is 0.022, KVI = 4.7, KMR = 1.05, without dual-detection maps fusion |
| | (VI)CFAR | The preset false alarm probability is 0.017, KVI = 4.7, KMR = 1.05 |
| | (CA)CFAR | The preset false alarm probability is 0.027 |

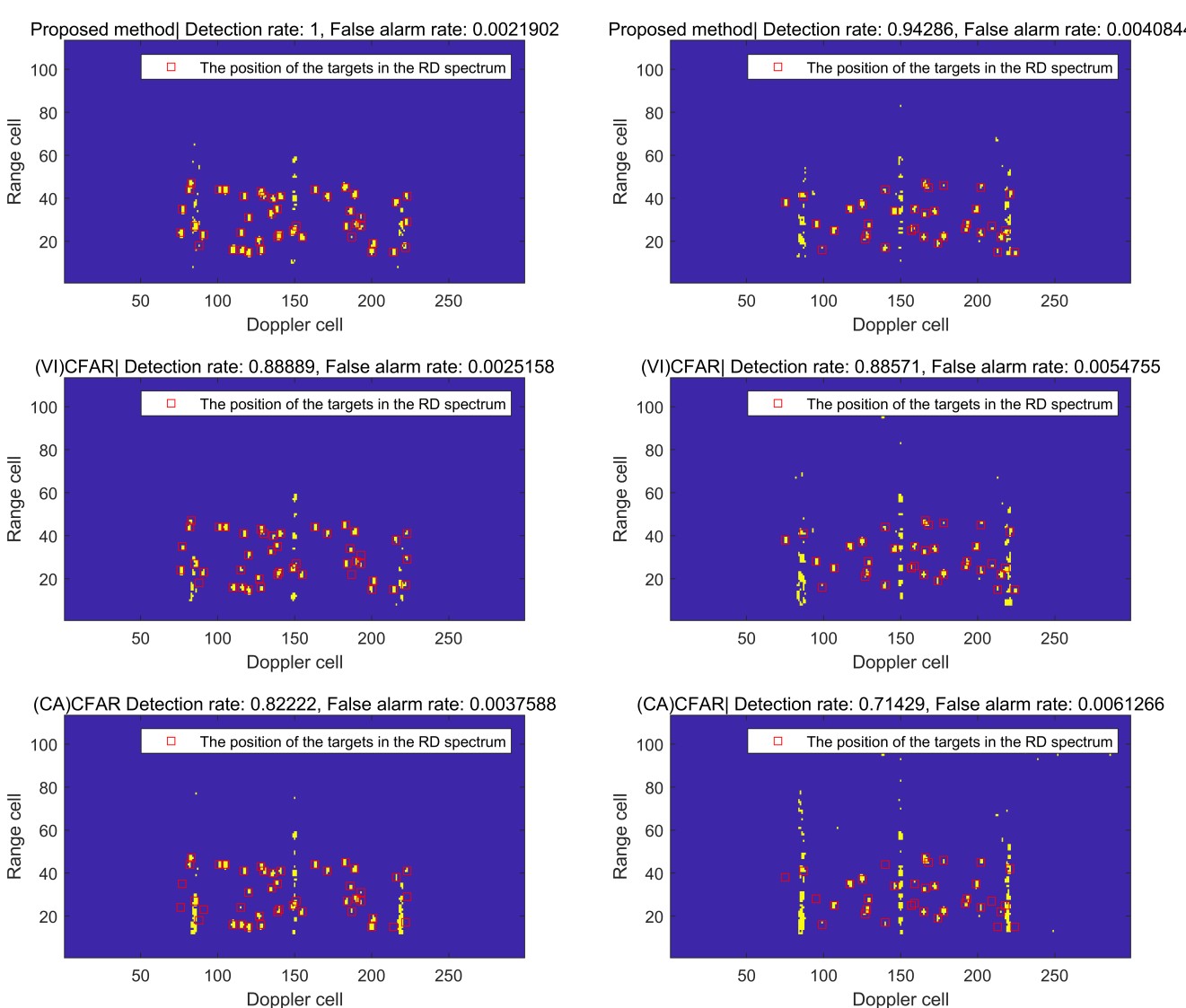

**Figure 18.** Comparison of the detection effects between the proposed method and traditional CFAR.

After comparing the detection effects of different methods, it can be found that the method in this paper can effectively mitigates false alarms and maintains the detection rate at a high level. When compared to (VI) and (CA)CFAR using the same frame of RD spectrum data, the proposed method demonstrates a superior performance in terms of both FAR and DR.

### 4.3.2. Contrast the Proposed Method against Other Machine Learning-Based Detection Methods

In this section, we present a comparison between the detection effects of the method proposed in this article and other radar target detection methods based on machine learning.

Similarly, we selected one frame of RD spectrum data from DATA_1 and DATA_2 as examples and employed various methods to assess their detection performance, as illustrated in Figure 19. The left column represents the detection effect using DATA_1 single frame RD spectrum data, while the right column represents the detection effect using the DATA_2 single frame RD spectrum data.

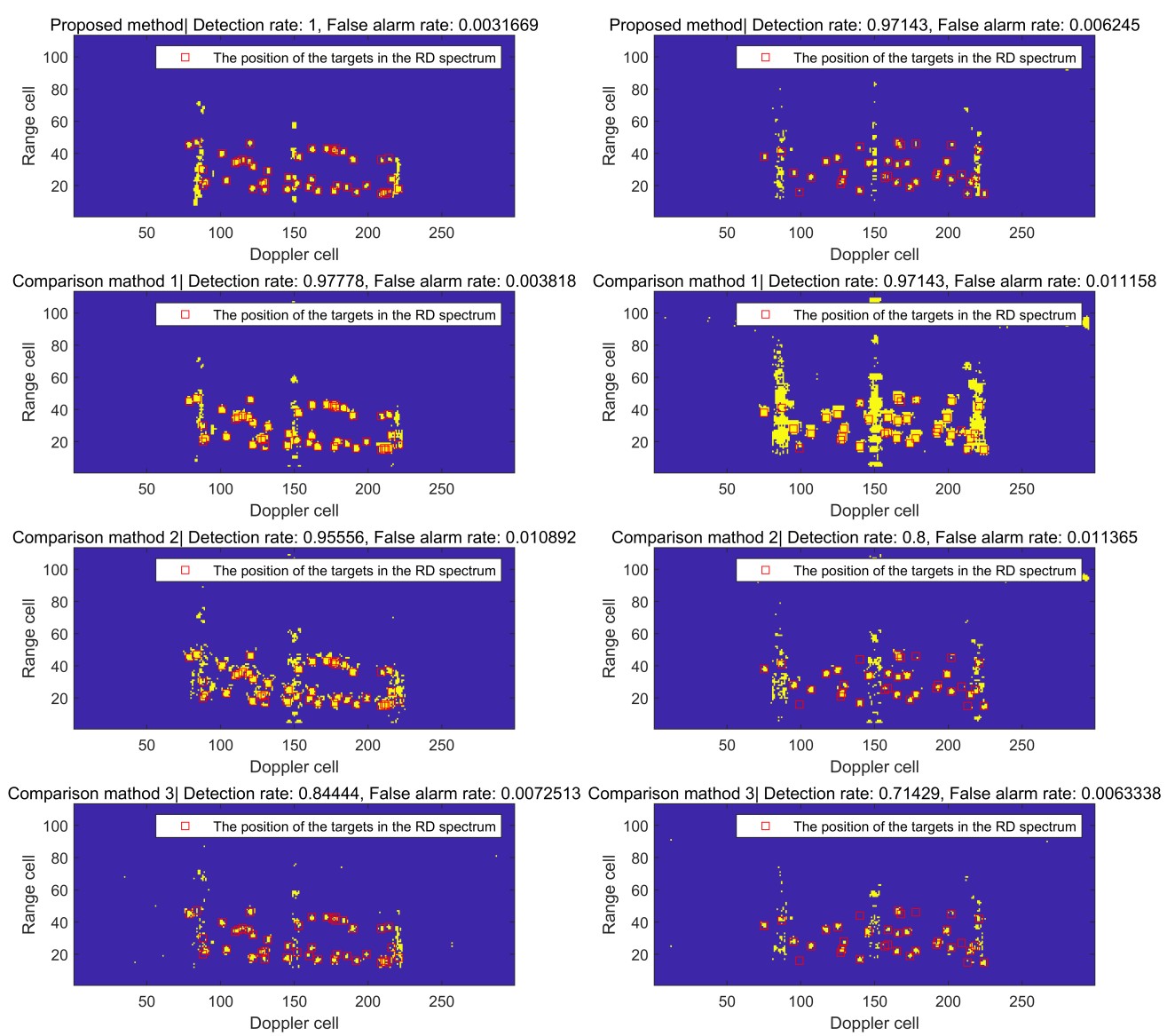

**Figure 19.** Comparison of the detection effects between the proposed method and other machine learning detection methods.

We compared three additional methods in this part, and provide the basic framework and references for each method in Table 7.

**Table 7.** Ideas and parameter settings of different methods.

| Detection Method | Method Ideas or Parameter Settings | References |
|---|---|---|
| Comparison method 1 | Sliding window+CNN | [16] |
| Comparison method 2 | (CA)CFAR supervised training, sliding window + ANN | [31] |
| Comparison method 3 | After the first-level detection, the window slice data features are extracted and sent to the classifier | [18] |
| Proposed method | (VI)CFAR-CNN is used as detector 1, with a preset false alarm probability of 0.07, KVI = 4.7, KMR = 1.05; (VI)CFAR is used as detector 2, with a preset false alarm probability of 0.08; a dual-detection maps fusion decision | - |

Among these methods, Comparison method 1 is referenced from the literature [16]. Initially, Doppler processing is conducted on the echo signal, and the sliding window

method is employed to input the window slice data into the CNN network for target detection. In fact, the CNN is utilized as a substitute for the CFAR algorithm.

Comparison method 2 refers to the method in the literature [31]. It involves employing the sliding window method to input both the detection unit and the reference unit into an artificial neural network, and determine whether there is a target in the detection unit through the network. In this process, (CA)CFAR is utilized to supervise the training of the artificial neural network.

Comparison method 3 refers to the fundamental framework outlined in the literature [18]. Initially, the morphological detection operator is employed on the RD spectrum to conduct first-level detection. Subsequently, a window is used to intercept the detection outcomes and extract pertinent features, which are then sent to the classifier for classification. To bolster the comparison method's generalizability and robustness, a support vector machine (SVM) is utilized as the classifier.

After comparing the detection effects of various methods, the proposed method in this article demonstrates a superior performance with the lowest FAR and the highest DR.

### 4.3.3. Comparison of Overall Performance of Multiple Methods

In this section, we selected 80 frames of the RD spectrum in DATA_1 and DATA_2 as test data, used the proposed method and several methods mentioned before for detection, and finally calculated the detection rate (DR), false alarm rate (FAR) of the missed detection rate (MR) and error rate (ER), where $MR = 1 - DR$ and $ER = FAR + MR$. The results are shown in Table 8.

**Table 8.** Comparison of the performance indicators of different methods.

| Detection Method | Performance Index | | | |
|:---:|:---:|:---:|:---:|:---:|
| | DR | FAR | MR | ER |
| **DATA_1** | | | | |
| (VI)CFAR | 0.963889 | 0.006920 | 0.036111 | 0.043031 |
| (CA)CFAR | 0.954722 | 0.008780 | 0.045278 | 0.054058 |
| Comparison method 1 | 0.959420 | 0.006615 | 0.040580 | 0.047195 |
| Comparison method 2 | 0.935667 | 0.016068 | 0.119174 | 0.080401 |
| Comparison method 3 | 0.880826 | 0.007218 | 0.064333 | 0.126392 |
| **Proposed method** | **0.978667** | **0.004945** | **0.021333** | **0.026278** |
| **DATA_2** | | | | |
| (VI)CFAR | 0.929643 | 0.010523 | 0.070357 | 0.080880 |
| (CA)CFAR | 0.921786 | 0.011277 | 0.078214 | 0.089491 |
| Comparison method 1 | 0.937500 | 0.013386 | 0.062500 | 0.075886 |
| Comparison method 2 | 0.908571 | 0.018026 | 0.091429 | 0.109455 |
| Comparison method 3 | 0.866071 | 0.012123 | 0.133929 | 0.146052 |
| **Proposed method** | **0.946071** | **0.010078** | **0.053929** | **0.064007** |

The results indicate that the proposed method outperforms other methods in all performance indices on the two datasets: DATA_1, characterized by a higher SNR and SCR, and DATA_2, characterized by a lower SNR and SCR. Specifically, the detection rate is the highest, the false alarm rate is the lowest, and the missed alarm rate and error rate are also minimized.

### 4.4. Verification of the Measured Target Data

In previous experiments, we used the target embedding method to construct a large amount of data in order to more comprehensively and accurately evaluate the performance of proposed method. To assess the feasibility of proposed method in engineering applications, it is essential to test it using the real-world data of measured targets. In this

section, we correlated AIS data with HFSWR data, tracked the trajectory information of a cooperative ship, and conducted target calibration using three frames of RD spectra.

In the first frame of RD spectrum, the radial distance of the ship target is 62.77 km and the radial speed is −3.46 m/s. The method proposed in this paper was utilized to detect the RD of the measured target in this frame, and its detection effect was compared with that of (CA)CFAR. The RD spectrum containing the measured targets is shown in Figure 20. The detection results are presented in Figures 21 and 22.

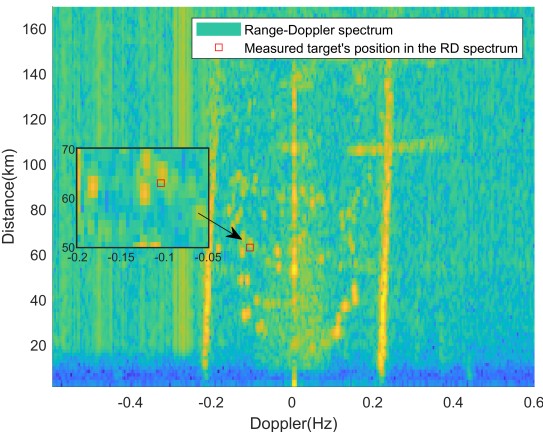

**Figure 20.** RD spectrum of the measured target in the first frame.

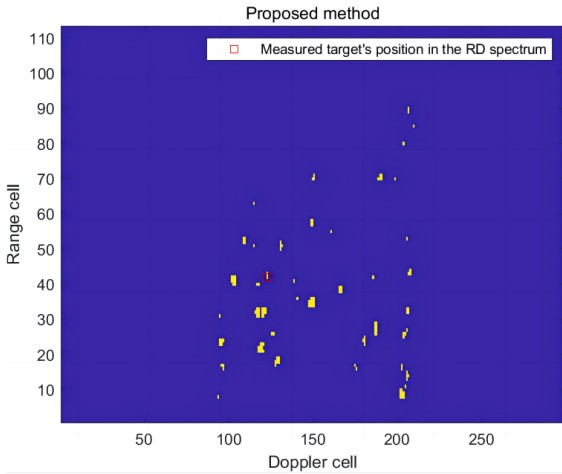

**Figure 21.** The detection effect of the RD spectrum in the first frame of this article's method.

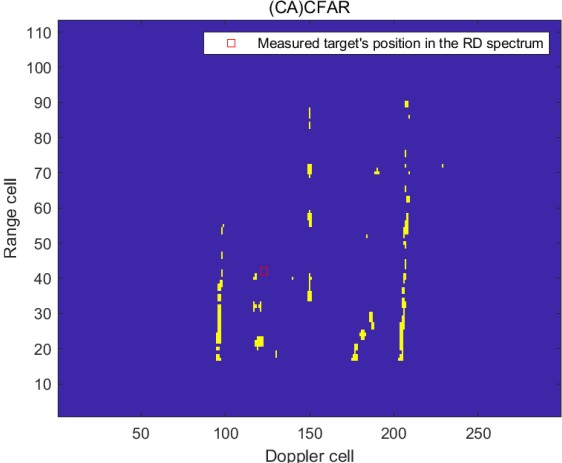

**Figure 22.** The detection effect of the RD spectrum in the first frame of (CA)CFAR.

In the second frame of the RD spectrum, the radial distance of the ship target is 68.7 km, and the radial speed is −3.79 m/s. The RD spectrum containing the measured targets is shown in Figure 23. The detection results are presented in Figures 24 and 25.

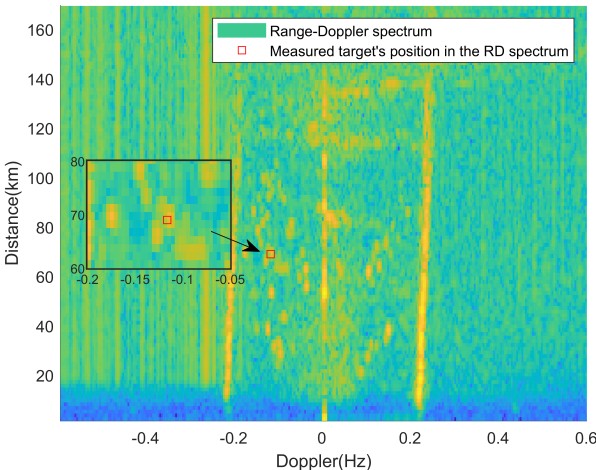

**Figure 23.** RD spectrum of the measured target in the second frame.

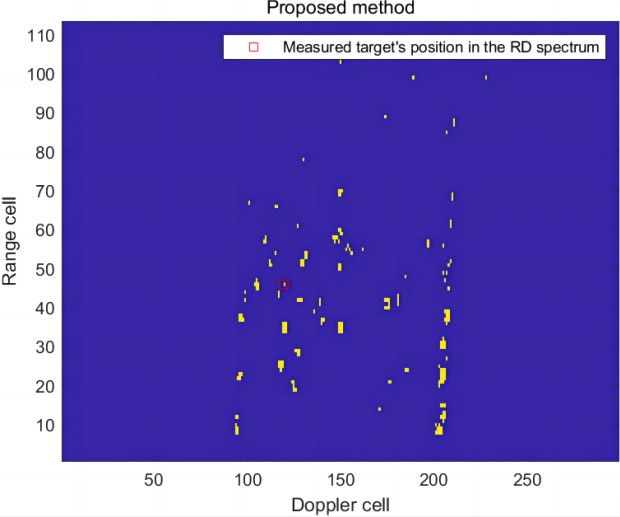

**Figure 24.** The detection effect of the RD spectrum in the second frame of this article's method.

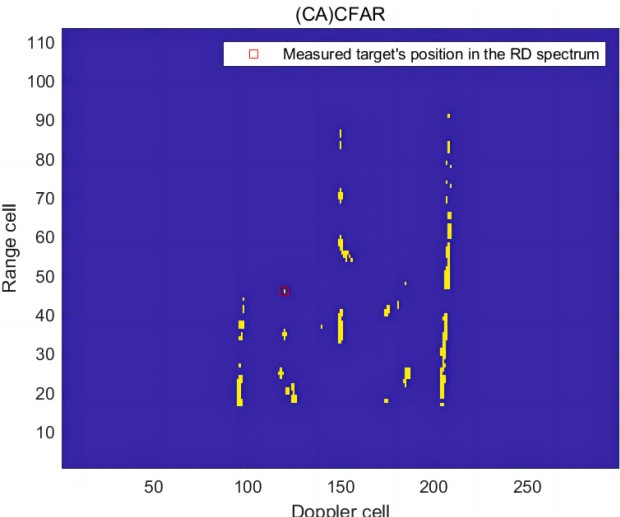

**Figure 25.** The detection effect of the RD spectrum in the second frame of (CA)CFAR.

In the third frame of RD spectrum, the radial distance of the ship target is 67.5 km, and the radial speed is $-3.80$ m/s. The RD spectrum containing the measured targets is shown in Figure 26. The detection results are presented in Figures 27 and 28.

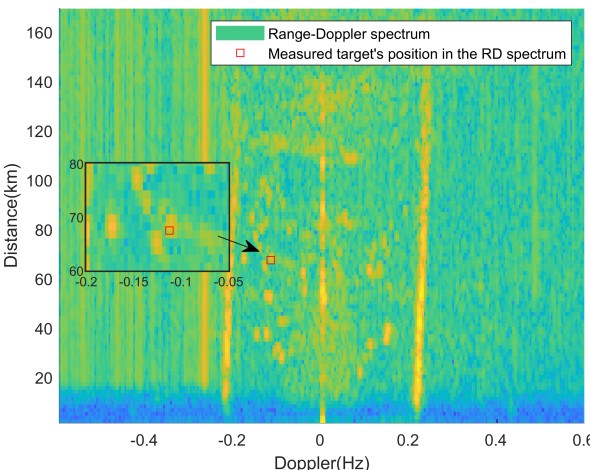

**Figure 26.** RD spectrum of the measured target in the third frame.

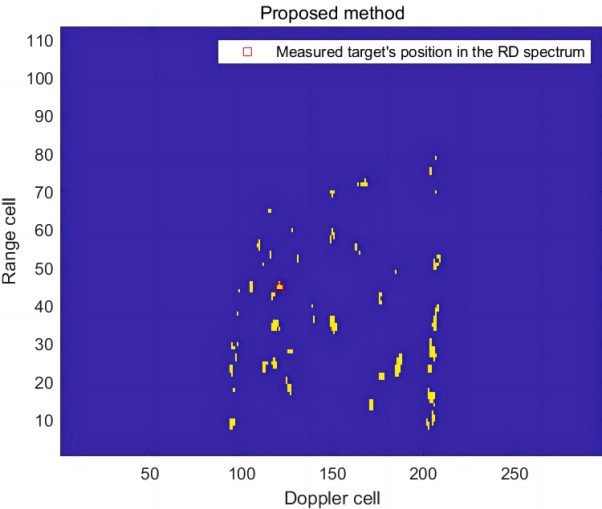

**Figure 27.** The detection effect of the RD spectrum in the third frame of this article's method.

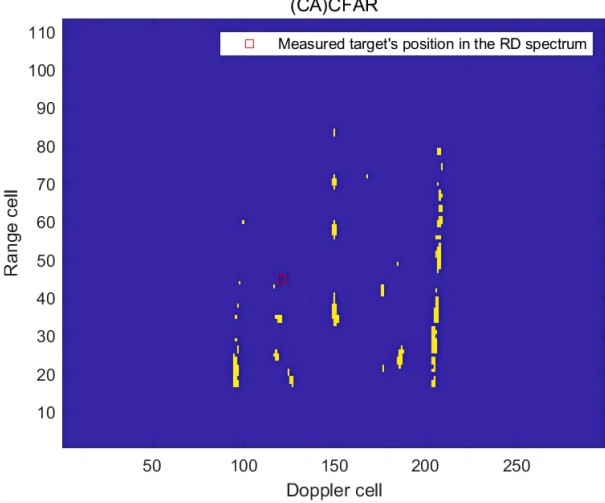

**Figure 28.** The detection effect of the RD spectrum in the third frame of (CA)CFAR.

After analyzing the detection results of the three frames of measured target RD spectra, it is evident that the proposed method in this article accurately detects targets in the complete measured RD spectrum. This is achieved by training and building a detection model using simulation data. Furthermore, compared to the traditional CFAR method, this approach effectively mitigates false alarms caused by different clutter types.

## 5. Conclusions

Inspired by the effective role of human visual processing in target detection and false alarm control, this paper proposes the use of an adaptive constant false alarm detector (VI) CFAR and CNN network cascade detector for target detection on the RD spectrum of HFSWR. Addressing the problem of the limited detection performance of CFAR-CNN caused by the convergence of the CNN model, a method of fusion of dual-detection maps was further proposed to compensate for the loss of detection performance. Afterwards, datasets were constructed using the target embedding method, and the performance of the proposed method in this article was verified and evaluated. The experimental results demonstrate the effectiveness and superiority of the method. Additionally, the feasibility of applying this method in practical applications was confirmed using the target data obtained from complete measurements. In conclusion, the method proposed in this article holds a research value and offers a new direction for the application of machine learning in CFAR processing.

**Author Contributions:** Conceptualization, Y.J., A.L. and C.Y.; methodology, Y.J.; software, Y.J.; validation, Y.J.; formal analysis, Y.J.; investigation, Y.J.; resources, Y.J. and X.C.; data curation, Y.J., J.W. and X.C.; writing—original draft preparation, Y.J.; writing—review and editing, Y.J.; visualization, Y.J.; supervision, A.L. and C.Y.; project administration, A.L. and C.Y.; funding acquisition, A.L. and C.Y. All authors have read and agreed to the published version of the manuscript.

**Funding:** The research and publication of the article were funded by the National Natural Science Foundation of China under Grant 62031015 and Mount Taishan Scholar Distinguished Expert Plan under Grant 20190957.

**Data Availability Statement:** The data pertaining to this scientific research are currently unavailable for public access due to confidentiality constraints. For inquiries regarding access to the data, please get in touch with the corresponding author.

**Acknowledgments:** We extend our appreciation to the editor and anonymous reviewers for their valuable insights and recommendations. Additionally, our gratitude goes to the research team stationed at the Harbin Institute of Technology, Weihai, for generously sharing the high-frequency surface wave radar data with us.

**Conflicts of Interest:** The authors declare no conflicts of interest.

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
