# Peer review of "Target Detection Method for High-Frequency Surface Wave Radar RD Spectrum Based on (VI)CFAR-CNN and Dual-Detection Maps Fusion Compensation"

_remotesensing, doi:10.3390/rs16020332_

Round 1
Reviewer 1 Report
Comments and Suggestions for Authors
A good organized paper. The problem formulation and the proposed solutions are good with nice presentation of the results. The proposed method has novelty and also good contribution to scientific society.
Author Response
We express our gratitude to the Editor for their handling of our manuscript and giving us the opportunity to revise it. Additionally, we would like to thank the reviewers for their valuable comments which have greatly contributed to improving the manuscript. To acknowledge their input, we have included an acknowledgment section in the paper.
In the following, we present a point-by-point response to the review comments. To make it easier to read, the comments from each reviewer are in blue italics while our response are non-italicized. We hope you will find our new revisions satisfactory.
All the comments of editor and reviewers with our responses are given as follows.
REVIEWER 1
A good organized paper. The problem formulation and the proposed solutions are good with nice presentation of the results. The proposed method has novelty and also good contribution to scientific society.
OUR RESPONSE: Thank you very much for your valuable feedback and positive evaluation of our paper. We’re delighted that you found the problem formulation, solutions, and result presentation satisfactory. Your recognition of the novelty and contribution of our proposed method to the scientific community is greatly appreciated. Thank you once again for your time and insightful review.

Reviewer 2 Report
Comments and Suggestions for Authors
In this paper, the authors propose a detection model that combines CFAR detector with a convolutional neural network (CNN), integrating image processing and computer vision techniques with CFAR processing. In general, the work in this paper is very interesting. However, the paper should be improved before acceptance.
1. The CFAR detector is a classical method. However, the authors just review few work about this method. The authors should comprehensively conduct literature review about this method in section 1.
2. In line 245 on page 9: The 9*9 rectangular window is used. The reviewer wanders to know how to determine the size of the window. Is there a method to determine the size? The authors should discuss the influence of window size on performance.
3. The optimization equation shown as Eq. (21) is used. The authors should discuss the robustness of this optimization method. Besides, the authors should also discuss the
4. In line 392 on page 14: The step 5 shows that ‘a vector consisting of three variables ω1, ω2 and μ’. The reviewer wanders to know how to set the initial values of these variables at first. The initial values of these parameters highly affect the robustness of optimization method.
5. In practice, the sidescan sonar can also provide high resolution image [1][2], which is helpful for the target detection. The reviewer wanders to know whether the authors’ method can be applied to sidescan sonar. The authors should discuss this issue in their paper, and the discussion would be helpful for the readers.
[1]X. Zhang,et al.Multireceiver SAS imagery based on monostatic conversion.IEEE Journal of Selected Topics in Applied Earth Observations and Remote Sensing,2021,14:10835-10853.
[2]Yang, P., An imaging algorithm for high-resolution imaging sonar system, Multimedia Tools and Applications, 2023, pp. 1-16, Doi: 10.1007/s11042-023-16757-0
5. The detection rate and false alarm rate highly depends on SNR. However, the simulation section does not discuss the performance versus with SNR. The authors should add this.
Author Response
We express our gratitude to the Editor for their handling of our manuscript and giving us the opportunity to revise it. Additionally, we would like to thank the reviewers for their valuable comments which have greatly contributed to improving the manuscript. To acknowledge their input, we have included an acknowledgment section in the paper.
In the following, we present a point-by-point response to the review comments. To make it easier to read, the comments from each reviewer are in blue italics while our response are non-italicized. We hope you will find our new revisions satisfactory. All comments from editors and reviewers, as well as details of our responses, can be found in the uploaded PDF file.
REVIEWER 2
In this paper, the authors propose a detection model that combines CFAR detector with a convolutional neural network (CNN), integrating image processing and computer vision techniques with CFAR processing. In general, the work in this paper is very interesting. However, the paper should be improved before acceptance.
OUR RESPONSE: Thank you for your valuable review and acknowledgment of our work.
We appreciate your recognition of the work’s overall interest. We value your feedback and will diligently work on enhancing the paper based on your suggestions to ensure its suitability for acceptance. Your insights are incredibly valuable to us, and we will address the areas that require improvement meticulously.
Once again, we sincerely appreciate your valuable time and detailed assessment of our work.

Reviewer 3 Report
Comments and Suggestions for Authors
Author Response
We express our gratitude to the Editor for their handling of our manuscript and giving us the opportunity to revise it. Additionally, we would like to thank the reviewers for their valuable comments which have greatly contributed to improving the manuscript. To acknowledge their input, we have included an acknowledgment section in the paper.
In the following, we present a point-by-point response to the review comments. To make it easier to read, the comments from each reviewer are in blue italics while our response are non-italicized. We hope you will find our new revisions satisfactory.
All comments from editors and reviewers, as well as details of our responses, can be found in the uploaded PDF file.
REVIEWER 3
In this paper, the authors present a method for intelligent detection of HFSWR targets. The paper is theoretically well-based, and the mathematical model is correct. I think also that the method is presented in an original way, but with certain modifications it is possible to make the paper of even better quality. I think some improvements should be made for better understanding and interest to readers. So, it made me suggest this manuscript to Reconsider after major revision.
OUR RESPONSE: Thank you for conducting a comprehensive evaluation of our work.
We are delighted to hear your recognition of the solid theoretical foundation and accurate mathematical models presented in our article. We highly value your constructive feedback and agree that improving the quality of the paper is crucial for enhancing reader understanding. We have carefully considered your suggestions and made significant improvements to enhance the appeal of the manuscript.
Once again, we sincerely appreciate your valuable time and detailed assessment of our work.

Round 2
Reviewer 2 Report
Comments and Suggestions for Authors
The paper is improved after revision and it can be accepted.
Reviewer 3 Report
Comments and Suggestions for Authors
From the aspect of my knowledge and experience, the work is good, interesting and related to the actual issue of OTHR radars
best regards
Reviewer